# Selection of Autochthonous LAB Strains of Unripe Green Tomato towards the Production of Highly Nutritious Lacto-Fermented Ingredients

**DOI:** 10.3390/foods10122916

**Published:** 2021-11-25

**Authors:** Nelson Pereira, Carla Alegria, Cristina Aleixo, Paula Martins, Elsa M. Gonçalves, Marta Abreu

**Affiliations:** 1Departamento de Química, Faculdade de Ciências e Tecnologia, Universidade Nova de Lisboa, 2829-516 Caparica, Portugal; nmv.pereira@campus.fct.unl.pt; 2Unidade de Tecnologia e Inovação, Instituto Nacional de Investigação Agrária e Veterinária, 2780-157 Oeiras, Portugal; cristina.aleixo@iniav.pt (C.A.); paula.martins@iniav.pt (P.M.); 3SFCOLAB—Associação Smart Farm COLAB Laboratório Colaborativo para a Inovação Digital na Agricultura, Rua Cândido dos Reis nº1, Espaço SFCOLAB, 2560-312 Torres Vedras, Portugal; carla.alegria@sfcolab.org; 4cE3c—Centre for Ecology, Evolution and Environmental Changes, Faculdade de Ciências, Universidade de Lisboa, Ed. C2, 1749-016 Lisboa, Portugal; 5GeoBioTec—Geobiociências, Geoengenharias e Geotecnologias, Faculdade de Ciências e Tecnologia, Universidade Nova de Lisboa, 2829-516 Caparica, Portugal; 6LEAF (Linking Landscape Environment Agriculture and Food) Research Centre, School of Agronomy, University of Lisbon, Tapada da Ajuda, 1349-017 Lisboa, Portugal

**Keywords:** *Lactiplantibacillus plantarum*, *Weissella paramesenteroides*, molecular identification, probiotic potential, solanine, industrial-crop-waste valorisation

## Abstract

Lactic fermentation of unripe green tomatoes as a tool to produce food ingredients is a viable alternative for adding value to industrial tomatoes unsuitable for processing and left in large quantities in the fields. Fermentation using starter cultures isolated from the fruit (plant-matrix adapted) can have advantages over allochthonous strains in obtaining fermented products with sensory acceptability and potentially probiotic characteristics. This paper details the characterisation of the unripe green tomato lactic microbiota to screen LAB strains for use as starter cultures in fermentation processes, along with LAB strains available from INIAV’s collection. Morphological, biochemical (API system), and genomic (16S rDNA gene sequencing) identification showed that the dominant LAB genera in unripe green tomato are *Lactiplantibacillus*, *Leuconostoc*, and *Weissella*. Among nine tested strains, autochthonous *Lactiplantibacillus plantarum* and allochthonous *Weissella paramesenteroides* showed tolerance to added solanine (200 ppm) and the best in vitro probiotic potential. The results indicate that the two LAB strains are promising candidates for manufacturing probiotic fermented foods from unripe green tomatoes.

## 1. Introduction

In today’s era of eco-security and eco-preservation, the world’s growing concern about food losses warrants urgent attention. Among the different food sectors, fruits and vegetables represent a significant source of waste and by-products, estimated to be around 40–50% per year worldwide [1]. Several studies have been conducted on waste and by-product valorisation (reducing and recycling) to mitigate food losses in the fruit and vegetable value chain [2,3]. Innovative valorisation strategies have emerged, including the recovery of high-value-added ingredients from plant materials [2,4].

On a global scale, the annual production of fresh tomatoes amounts to approximately 180 million tons, with a contribution of 17 million tons from Europe [5]. The wasted agricultural biomass in the tomato value chain thus has a vast potential for recovery. It includes fractions considered unsuitable for sale in the fresh fruit market (colour, shape, ripeness, unacceptable injuries) and side flows from the processing industry [6].

The main value-added approaches adopted for the valorisation of tomato by-products and their components mainly include animal feed, functional ingredients for food products, and raw materials for nutraceuticals [4]. In Portugal, the tomato industry focuses on obtaining a single high-value product—tomato paste, and only fruits at the red ripe stage enter the processing lines. The amount of unripe green tomatoes left in the fields depends on the harvest year and may reach about 1.12 × 10^8^ kg/year [7]. The immature fruits represent high economic losses for producers and a negative environmental impact.

Tomato (*Lycopersicon esculentum* L.) is a valuable source of vital nutrients such as minerals, dietary fibres, vitamins, essential amino acids, sugars, carotenoids, and phenolics [8]. Green tomato composition differs from that of mature red tomato: unripe fruit has lower amounts of sugars, lycopene, and ascorbic acid, regardless of the cultivar [9]. Nevertheless, the content of tomato glycoalkaloids (SGAs), namely α-tomatin and dehydrotomatin, is higher in unripe than in ripe tomatoes (500 vs. 5 mg.kg^−1^, respectively) [10,11]. Food valorisation of unripe green tomatoes must consider this potential health risk [12] for SGA levels above the 200 ppm limit [13].

In recent years there has been considerable research into the utilisation of by-products and wastes through fermentative processes using lactic acid bacteria (LAB) and other microorganisms for manufacturing a wide variety of fermented foods [14,15]. Fermentation can be carried out by natural flora or by inoculated starters. However, starter cultures improve the quality of fermented foods by providing better fermentation control and predictability of the final product [16]. For example, Simões et al. [7] studied the fermentation of unripe tomato using a consortium of yeast and lactic acid bacteria, obtaining an ingredient of high nutritional and sensory quality with potential uses in the development of new food products (e.g., salad dressings and sauces).

Many authors have explored the potential use of autochthonous bacterial strains in the fermentation process, given that they have an advantage over allochthonous strains in being more niche-specific [17,18]. The selection of LAB strains from raw tomatoes has also proved essential for developing fermented products with a suitable volatile profile that positively affects their flavour and aroma [19,20]. LAB strains are the most widely used microorganisms for starter cultures for developing functional foods because they are generally recognised as safe (GRAS) and have the potential to reduce the content of anti-nutrients during the fermentation process [21]. As an example, the use of appropriate LAB strains provides an opportunity to mitigate the potential risk of high glycoalkaloid content in fermented products: some *Lactobacilli* spp. have been shown to reduce SGAs by up to 50% [22,23,24]. On the other hand, Friedman [25] suggests an inhibitory effect of SGAs on *Lactobacilli* spp., which might affect the fermentative potential of lactic strains. 

Developing healthy products based on probiotic fermented fruit and vegetables is another key research area for the future functional food market. Despite the array of LAB starters available for dairy, meat, and baked goods fermentation, only a few show suitability for vegetable fermentation [26], and even fewer have recognised probiotic potential [27].

Despite numerous papers published on the effects of inoculums on tomato fermentation products, to the best of our knowledge, none are dedicated to a systematic selection of strains isolated from unripe green tomato lactic microbiota. Finding multifunctional LAB strains from autochthonous green tomato microbiota is highly advantageous. This study aims to identify autochthonous strains isolated from unripe green tomato lactic microbiota to select the most promising strains as starter cultures for developing probiotic high-value lacto-fermented products.

## 2. Materials and Methods

### 2.1. Plant Material

Unripe green industrial tomato (stage 1 fruit according to the USDA colour classification requirements) from the variety H1015 was harvested from random rows of multiple plantations distributed over the Ribatejo region of Portugal (municipalities of Azambuja, Benavente, Cartaxo, and Vila Franca de Xira). Upon arrival at the laboratory, undamaged unripe green tomatoes were selected, frozen, and kept at −20 °C (Cryocell Aralab, Rio de Mouro, Portugal) until analysis. 

### 2.2. Isolation, Characterisation, and Identification of Autochthonous LAB

To identify and characterise potential functional starter cultures from unripe green tomato microbiota, fruit samples were blended with tryptone salt broth and serially diluted (up to 10^−6^) in tryptone salt broth (Biokar Diagnostics, Allonne, France). One millilitre of each dilution was pour-plated in de Man Rogosa and Sharpe (MRS) agar (Biokar Diagnostics, Allonne, France) and incubated at 30 °C for 72 h under anaerobic conditions (ISO 15214:1998). To obtain pure cultures, about 5–10 representative colonies from each plate were phenotypically selected and sub-cultured on MRS broth (Biokar Diagnostics, Allonne, France). The stock cultures were kept at −80 °C in MRS broth supplemented with 50% (*v/v*) glycerol (Sigma-Aldrich, St. Louis, MO, USA) as cryoprotectant, until further analysis.

The LAB’s morphological and biochemical characteristics were evaluated after 24 h incubation on MRS agar and characterised by microscopic morphological examination (Leitz Dialux 20, Leica Microsystems, Wetzlar, Germany), catalase activity, and KOH test. Biochemical identification of LAB isolates was obtained using API 50 CHL kit (BioMérieux S.A, Marcy L′Étoile, France) procedures according to the manufacturer’s specifications, and the results were analysed using API-web software (BioMérieux V 5.1, Marcy L′Étoile, France).

### 2.3. DNA Extraction for Molecular Identification 

Seven autochthonous LAB strains from unripe green tomato were selected based on isolation frequency (>10%) and relevance for green tomato spontaneous fermentation (ID: LAB40, LAB49, LAB67, LAB82, LAB89, LAB94, and LAB97) and further identified by 16S rDNA gene sequencing. 

A preliminary study screened several fermentative LAB strains from the INIAV bacteria strain collection to grow and ferment unripe green tomato as a substrate medium. The screening proved that two strains, C244 and C1090, had high microbial growth and sharp pH increments compared to other strains [28]. Therefore, these LAB strains were also tested along with selected unripe green tomato autochthonous strains.

For the genomic analysis, each colony was suspended in 10 µL of sterile phosphate-buffered saline pH 7.4 (PBS; Sigma-Aldrich, St. Louis, MO, USA) and then applied to an FTA Micro Card (Flinders Technology Associates), according to the in-house procedure of an independent sequencing service, StabVida (Lisbon, Portugal). The samples were dried for two hours at room temperature, stored individually in plastic bags, and sent to StabVida. Sanger sequencing of the 16S rRNA was performed using the universal primer sets 27F/1492R and 518F/800R. The obtained sequences were compared with the publicly available National Center for Biotechnology Information 16S rRNA database (NCBI, http://www.ncbi.nih.gov, accessed on 9 November 2021)) using the advanced BLAST search tool (http://blast.ncbi.nlm.nih.gov/Blast.cgi, accessed on 9 November 2021). 

### 2.4. In Vitro Probiotic Potential of Selected Lab Strains 

Two culture media were used for LAB growth assessment under artificial gastric conditions: MRS broth and unripe green tomato mixed with distilled water (1:1, *w/w*), identified as MRS and GT, respectively. 

The strains’ resistance to a low pH environment mimicking gastric conditions was assessed by adjusting the MRS and GT media to pH 2.5 (0.1 M HCl). The media were then inoculated (8 log_10_ CFU/mL) and incubated at 37 °C for 3 h. For assessing bile salt tolerance, LAB strains were inoculated into the MRS and GT media containing 0.3% bile salt mixture (Sigma-Aldrich, St. Louis, MO, USA) and incubated at 37 °C for 4 h. Controls were set up following the same procedure without pH correction and bile salt addition. 

The viable colony counts (log_10_ CFU/mL) from each test condition were determined using the pour-plate method (MRS agar) with the appropriate decimal dilutions after incubation at 30 °C for 72 h [29], and survival rates (%) were estimated (N_f_/N_0_ × 100%, with N_0_ as the initial count at time 0 h and N_f_ as the final LAB count after the incubation period). All assays were executed in triplicate.

### 2.5. Selected LAB Strains’ Tolerance to Solanine 

The glycoalkaloid inhibitory effect on LAB growth [25] was evaluated by adding 200 ppm of solanine (Sigma-Aldrich, St. Louis, MO, USA) to the MRS broth further inoculated with each of the selected LAB strains (9 log_10_ CFU/mL) and incubated at 30 °C for 72 h (in triplicate). Viable cell counts (log_10_ CFU/mL) [29] were determined at 0 h and after 24 h, 48 h, and 72 h of incubation. 

### 2.6. Statistical Analysis

The statistical analysis was performed using Statistica^TM^ V8.0 software [30]. Factorial ANOVA was applied, followed by Tukey’s HSD test, to detect significant differences (*p* < 0.05) in the data regarding tolerance to in vitro simulated GI tract conditions and LAB growth at threshold solanine concentrations.

## 3. Results and Discussion

### 3.1. Isolation, Characterisation, and Identification of Autochthonous LAB

Fifty-five (55) LAB strains were isolated from the unripe green tomato microbiota based on distinct and observable phenotypic characteristics. Regarding the microscopic observations (shape, size, mobility, and aggregation), most isolated strains showed a rod shape (≈75%), and only 25% were identified as *Lactococcus*. Irrespective of shape, all strains differed in width, size, and aggregation. From the biochemical standpoint, all isolated strains tested as catalase-negative and Gram-positive, corresponding to LAB characteristic profiles. Moreover, the API system confirmed the identification of most isolates.

The API method revealed different carbohydrate fermentation profiles, indicating diversity among the unripe green tomato autochthonous lactic microbiota, with over 90% of isolated strains fermenting D-glucose, D-fructose, D-mannose, N-acetyl-glucosamine, esculin, and cellobiose. Furthermore, species identification was achieved through API-web software, providing identity matches between 52.9 and 99.9% (Table 1): of the 55 isolates, 14 were poorly discriminated (species identification < 80%), 4 showed similarities between 80 and 90%, 21 between 90 and 99%, 2 ranging from 99 to 99.9%, and finally 14 isolates had a similarity ≥ 99.9%.

The morphological and biochemical characterisation clearly revealed the unripe green tomato microbiota to be predominantly composed of lactic acid bacteria (Figure 1), with *Leuconostoc* (31%), *Lactobacillus (28%)*, and *Weissella* (25%) as the predominant autochthonous LAB genera.

Various studies support the identified genera prevalence in tomato products such as spontaneously fermented green tomatoes (*L. mesenteroides*, *L. casei*, *L. citreum*, *W. confusa*, *L. lactis,* and *L. plantarum*) [31,32], fresh tomato juice (*L. plantarum*, *W. cibaria* and *W. confusa*, *L. brevis* and *P. pentosaceus*) [17] and canned tomato (*Leuconostoc* sp., *Pediococcus* sp. and *Lactobacillus* sp.) [33]. 

Further molecular identification was carried out in the predominant LAB strains (>10% isolation frequency; Figure 1) and in the strains identified as *Lactococcus lactis* and *Lactobacillus plantarum*, due to their widespread use in the food industry as probiotic microorganisms and/or microbial starters [26]. Among the isolates identified as of the same species, those with higher identification percentages were chosen. In cases where the identification percentage matched between the same species, isolates were randomly selected. In addition, two LAB strains with proved ability to ferment unripe green tomato belonging to INIAV’s collection were also molecularly identified. 

### 3.2. Molecular Identification of Selected LAB Strains

To confirm the species and after checking for correct amplification of PCR products via 1.5% agarose gel electrophoresis, partial 16S rRNA gene sequences (approximately 1465 bp) of all isolates were determined and compared with related bacteria using the BLAST program at NCBI (http://blast.ncbi.nlm.nih.gov/Blast.cgi, accessed on 9 November 2021). The 16S rDNA gene sequencing allowed the identification of the selected strains with reliability higher than 99.9% (Table 2).

Four LAB genera were identified (*Lactiplantibacillus* (=*Lactobacillus*), *Weissella*, *Leuconostoc*, and *Lactococcus*) containing six different species, namely, *Lactiplantibacillus plantarum* (three isolates: LAB49, LAB97, and C244), *Weissella paramesenteroides* (one isolate: C1090), *Weissella cibaria* (two isolates: LAB40 and LAB89), *Weissella confusa* (one isolate: LAB94), *Leuconostoc citreum* (one isolate: LAB82), and *Lactococcus lactis* (one isolate: LAB67). The *Weissella paramesenteroides* species (C1090) was not found in the unripe green tomato isolates and originated from INIAV’s collection, as did one of the isolates identified as *Lactiplantibacillus plantarum* (C244).

The molecular identification was in good agreement with the results from the biochemical identification (API 50 CHL identification kit), with few exceptions. The strains C244 (*Lactiplantibacillus plantarum*), C1090 (*Weissella paramesenteroides*), LAB97 (*Lactiplantibacillus plantarum*), LAB82 (*Leuconostoc citreum*), and LAB67 (*Lactococcus lactis*) were identically identified by both methods. For the strain LAB89, both methods agreed on genus identification (*Weissella*) but differed in species identification (*W. cibaria* vs. *W. confusa*, for molecular and API identification, respectively). The identification of LAB40, LAB49, and LAB94 isolates diverged between molecular (Table 2) and API (Table 1) identification methods, possibly due to the purity of the isolated bacterial colonies. Other researchers have also observed discrepancies between the biochemical and genotypic data, e.g., [34,35]. The loss or gain of plasmids is considered a plausible justification since plasmids encode some genes needed for the fermentation of sugars, and variation in plasmid content may cause metabolic inconsistencies [35,36]. However, due to the stability of genomic DNA, whose sequences are not dependent on cultural conditions or handling, genotypic techniques are undoubtedly faster and more effective for species identification [34].

### 3.3. In Vitro Probiotic Potential of Selected LAB Strains

LAB strains need to withstand hostile gastric conditions (low pH and the presence of bile salts) and maintain a growth ≥7 log_10_ CFU/mL to have in vitro probiotic potential. Table 3 and Table 4 show the results of the different probiotic tests assayed according to the nine selected LAB genotypes in MRS and GT media, respectively. 

All strains reached counts > 7 log_10_ CFU/mL in control conditions (data not shown), maintaining high counts during the assessed period regardless of the medium. The difference between MRS and GT media counts did not exceed 1 log cycle, supporting the use of unripe green tomato as a culture medium, demonstrating the strains’ viability, and confirming that any inhibition of the strain growth was due to the limiting conditions of the test (pH 2.5, 0.3% bile salts).

Strain survival in both gastric pH and bile salts conditions differed between growth media, with the GT medium offering better conditions to support LAB growth and survival. Strain survival was negligible in MRS media and at pH 2.5 (Table 3), with only BAL89 reaching ca. 2 log_10_ CFU/mL counts after 3 h, even though all strains showed tolerance to bile salts (counts > 7 log_10_ CFU/mL). In GT medium (Table 4), LAB97 and C1090 were the strains with the highest overall survival (excluding controls with no limiting conditions, data not shown) in both low pH and bile salt mixture conditions (>95% and with counts > 7 log_10_ CFU/mL), while LAB67 and LAB82 were the strains most sensitive to in vitro digestion (0% survival). The remaining strains were sensitive to at least one of the tested conditions, showing differential tolerances to the other condition. 

Since we found significant differences in tolerance to digestive conditions among the tested strains, we regard the strains that showed significantly high tolerance as good candidates for further investigating the in vitro probiotic potential value (e.g., antibiotic susceptibility, antimicrobial activity, and bacterial adherence to stainless steel plates).

Probiotics must be able to survive the gastrointestinal tract transit, tolerating the highly acidic conditions in the stomach and the bile salts in the small intestine while maintaining their viability [37,38]. Therefore, in this preliminary screening, strains LAB97 (autochthonous *Lactiplantibacillus plantarum*) and C1090 (INIAV’s collection *Weissella paramesenteroides*) showed high in vitro probiotic potential. In fact, Sathyapriya and Anitha [39] and Paula et al. [40] showed that *Weissella paramesenteroides* exhibits high probiotic potential by tolerating low pH and bile salts in simulated gastrointestinal conditions. Likewise, *Lactiplantibacillus plantarum* is also one of the most studied species with proven probiotic potential [37,41,42].

### 3.4. Selected LAB Strains’ Tolerance to Solanine

The susceptibility to solanine of selected LAB strains was assessed. It has been shown that glycoalkaloids can inhibit LAB growth [25] and hence compromise the fermentation process. Nevertheless, it is reported that glycoalkaloids can be metabolised during the fermentation process [23,24], in which some types of *Lactobacillus* strains, including *Lactiplantibacillus plantarum*, can play a key role in diminishing the potential health risk posed [22,24]. Thus, considering the high amounts of glycoalkaloids present in unripe green tomatoes, it is hypothesised that differences between the ability of LAB strains to grow in the presence of solanine may provide some insight regarding their ability to degrade them. Figure 2 shows the LAB count for the selected nine LAB strains cultured in MRS, with and without added solanine (200 ppm), following the incubation period.

During incubation, no differences were found between LAB counts in MRS or MRS supplemented with solanine (*p* > 0.05), except for the LAB89 strain. All strains registered a LAB count increase (*p* < 0.05) after 24 h of incubation, followed by a decreasing trend for the remaining period, which may indicate that growth is unlikely to be limited by factors other than the nutrients in the MRS medium. LAB89 had lower counts in the solanine-enriched MRS medium after 48 and 72 h compared with in the absence of solanine; however, the differences between counts found in both media were below 1 log_10_ CFU/mL, making it hard to assess the strain’s sensitivity to solanine. Thus, in general, LAB growth was not influenced by the addition of solanine (200 ppm) despite some indication of differential tolerances of the selected strains to this glycoalkaloid. Given these results, we can speculate that the fermentation ability of the selected LAB strains (to be used as starters) will not be compromised by the high glycoalkaloid concentrations potentially present in unripe green tomato (up to 500 ppm). 

The glycoalkaloid degradation capacity of lactic fermentation can reach about 50% of the initial content [24] depending on the LAB strain [22,24]. It will be of the utmost interest to test the selected LAB strains in technological fermentation trials, including determination of the glycoalkaloid content, as a strategy to obtain safe products.

## 4. Conclusions

In the present study, 55 strains isolated from unripe green tomato were morphologically and biochemically identified. From these, seven LAB strains and two LAB strains from INIAV’s collection were selected and molecularly identified with the aim of evaluating their ability to drive unripe green tomato fermentation. Autochthonous *Lactiplantibacillus plantarum* LAB97 and allochthonous *Weissella paramesenteroides* C1090 from the INIAV collection tolerated high concentrations of solanine and were the only strains showing in vitro probiotic potential when cultured in unripe green tomato medium. Hence, these strains were found to be good candidates for further investigation as starters in laboratory and industrial fermentation, aiming to build a sound scientific background for highlighting the large potential of unripe green tomato, an industrial crop waste, for the development of new food ingredients.

## Figures and Tables

**Figure 1 foods-10-02916-f001:**
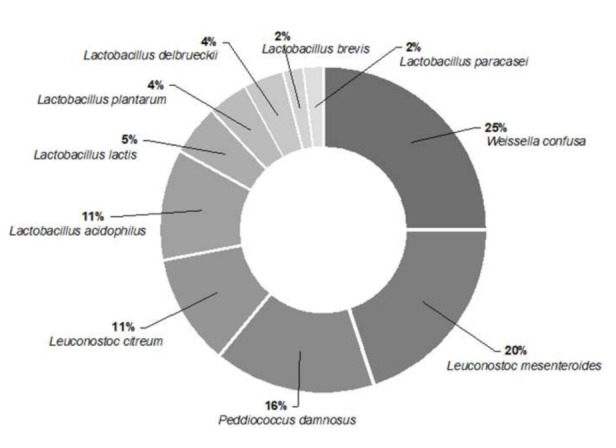
Autochthonous LAB species isolated from unripe green tomato (%).

**Figure 2 foods-10-02916-f002:**
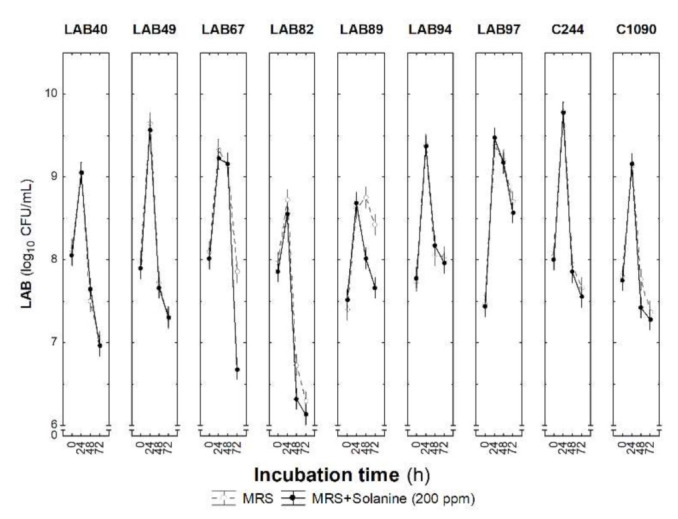
LAB counts (log_10_ CFU/mL) at 0, 24, 48, and 72 h of incubation in MRS broth, with and without solanine (200 ppm). Vertical bars show a 95% confidence interval (Tukey test at *p* = 0.05).

**Table 1 foods-10-02916-t001:** Species identification of unripe green tomato isolates based on carbohydrate fermentation profiles using API 50 CHL database.

Isolates	Species	ID%	Isolates	Species	ID%	Isolates	Species	ID%
LAB39	*Weissella confusa*	59.9	LAB58	*Pediococcus damnosus*	97.5	LAB81	*Lactobacillus acidophilus*	68.4
LAB40	*Leuconostoc mesenteroides*	99.9	LAB59	*Pediococcus damnosus*	97.5	LAB82	*Leuconostoc citreum*	99.9
LAB41	*Leuconostoc mesenteroides*	99.9	LAB60	*Leuconostoc mesenteroides*	90.4	LAB83	*Weissella confusa*	98.1
LAB42	*Weissella confusa*	99.6	LAB61	*Lacticaseibacillus paracasei*	52.9	LAB84	*Leuconostoc mesenteroides*	99.9
LAB43	*Weissella confusa*	96.6	LAB62	*Weissella confusa*	96.6	LAB85	*Leuconostoc citreum*	63.3
LAB44	*Lactobacillus brevis*	62.6	LAB63	*Weissella confusa*	96.6	LAB86	*Leuconostoc mesenteroides*	90.4
LAB45	*Lactococcus lactis*	73.9	LAB64	*Weissella confusa*	96.6	LAB87	*Weissella confusa*	99.6
LAB46	*Weissella confusa*	62.5	LAB65	*Weissella confusa*	96.6	LAB88	*Leuconostoc mesenteroides*	99.9
LAB47	*Weissella confusa*	83.4	LAB66	*Pediococcus damnosus*	97.5	LAB89	*Weissella confusa*	99.9
LAB48	*Weissella confusa*	83.4	LAB67	*Lactococcus lactis*	99.9	LAB90	*Lactobacillus acidophilus*	86.7
LAB49	*Pediococcus damnosus*	97.5	LAB68	*Lactococcus lactis*	99.9	LAB91	*Leuconostoc citreum*	87.7
LAB50	*Weissella confusa*	95	LAB69	*Leuconostoc mesenteroides*	97.5	LAB92	*Lactobacillus delbrueckii*	58.7
LAB51	*Leuconostoc mesenteroides*	97.8	LAB70	*Leuconostoc mesenteroides*	93.6	LAB93	*Lactobacillus acidophilus*	74.1
LAB52	*Pediococcus damnosus*	97.5	LAB75	*Leuconostoc citreum*	63.3	LAB94	*Lactobacillus acidophilus*	74.7
LAB53	*Leuconostoc mesenteroides*	99.9	LAB76	*Leuconostoc citreum*	99.9	LAB95	*Lactobacillus acidophilus*	74.1
LAB54	*Pediococcus damnosus*	97.5	LAB77	*Leuconostoc citreum*	99.9	LAB96	*Lactobacillus plantarum*	99.9
LAB55	*Pediococcus damnosus*	97.5	LAB78	*Lactobacillus delbrueckii*	58.7	LAB97	*Lactobacillus plantarum*	99.9
LAB56	*Pediococcus damnosus*	97.5	LAB79	*Leuconostoc mesenteroides*	99.9			
LAB57	*Pediococcus damnosus*	97.5	LAB80	*Lactobacillus acidophilus*	58.2			

ID%: identity (%), representing the similarities from the computer-aided database of the API-web software (API 50 CHL V5.1).

**Table 2 foods-10-02916-t002:** Suggested species of each consensus sequence BLASTed against NCBI nucleotide database.

Isolate Id.	Maximum Score	E-Value	Identity	Species (16S rRNA Gene Analysis)	Accession
LAB40	28378	0	100	*Weissella cibaria*	OL405452
LAB49	12941	0	100	*Lactiplantibacillus plantarum*	OL405451
LAB67	15019	0	100	*Lactococcus lactis*	OL405455
LAB82	10197	0	100	*Leuconostoc citreum*	OL405449
LAB89	41511	0	100	*Weissella cibaria*	OL405450
LAB94	2593	0	100	*Weissella confusa*	OL405453
LAB97	12849	0	100	*Lactiplantibacillus plantarum*	OL405454
C244 *	12858	0	99.93	*Lactiplantibacillus plantarum*	OL405447
C1090 *	20248	0	100	*Weissella paramesenteroides*	OL405448

* Strains from INIAV’s collection. Query coverage was 100%, and the maximum score matched the total score for all strains.

**Table 3 foods-10-02916-t003:** Survival rate and probiotic potential of LAB strains under simulated in vitro gastrointestinal conditions in MRS medium.

Strain	Species	Origin	Condition	Initial Count (Mean ± SD)	Final Count (Mean ± SD)	Survival Rate (%)	Probiotic Potential *
LAB40	*Weissella cibaria*	Autochthonous	pH 2.5	1.9 ± 0.3	0.0 ± 0.0	0	-
			0.3% bile salts	7.7 ± 0.1	7.7 ± 0.1	101	+
LAB49	*Lactiplantibacillus plantarum*	Autochthonous	pH 2.5	9.6 ± 0.0	0.0 ± 0.0	0	-
			0.3% bile salts	9.5 ± 0.0 *	8.8 ± 0.1	93	+
LAB67	*Lactococcus lactis*	Autochthonous	pH 2.5	7.2 ± 0.1	0.0 ± 0.0	0	-
			0.3% bile salts	7.0 ± 0.0	9.5 ± 0.0	135	+
LAB82	*Leuconostoc citreum*	Autochthonous	pH 2.5	5.9 ± 0.2	0.0 ± 0.0	0	-
			0.3% bile salts	9.4 ± 0.3	7.9 ± 0.1	84	+
LAB89	*Weisella cibaria*	Autochthonous	pH 2.5	6.6 ± 0.2	2.2 ± 0.0	34	-
			0.3% bile salts	7.7 ± 0.1	9.5 ± 0.0	123	+
LAB94	*Weissella confusa*	Autochthonous	pH 2.5	7.3 ± 0.0	0.0 ± 0.0	0	-
			0.3% bile salts	7.7 ± 0.1	8.2 ± 0.2	106	+
LAB97	*Lactiplantibacillus plantarum*	Autochthonous	pH 2.5	7.3 ± 0.0	0.0 ± 0.0	0	-
			0.3% bile salts	7.7 ± 0.0	8.3 ± 0.0	109	+
C244	*Lactiplantibacillus plantarum*	INIAV’s collection	pH 2.5	7.0 ± 0.3	0.0 ± 0.0	0	-
			0.3% bile salts	8.0 ± 0.2	7.2 ± 0.4	91	+
C1090	*Weissella paramesenteroides*	INIAV’s collection	pH 2.5	7.3 ± 0.5	0.0 ± 0.0	0	-
			0.3% bile salts	7.9 ± 0.0	8.7 ± 0.0	110	+

(+) LAB strain meets the minimal cell amount (10^7^ CFU/mL) to have probiotic potential; (-) LAB strain does not meet the minimal cell amount (10^7^ CFU/mL) to have probiotic potential; * (+) LAB strain meets the minimal cell amount (107 CFU/mL) to have probiotic potential; (-) LAB strain does not meet the minimal cell amount (107 CFU/mL) to have probiotic potential.

**Table 4 foods-10-02916-t004:** Survival rate and probiotic potential of LAB strains under simulated in vitro gastrointestinal conditions in unripe green tomato (GT) medium.

Strain	Species	Origin	Condition	Initial Count (Mean ± SD)	Final Count (Mean ± SD)	Survival Rate (%)	Probiotic Potential *
LAB40	*Weissella cibaria*	Autochthonous	pH 2.5	3.5 ± 0.1	3.3 ± 0.7	94	-
			0.3% bile salts	0.0 ± 0.0	0.0 ± 0.0	0	-
LAB49	*Lactiplantibacillus plantarum*	Autochthonous	pH 2.5	9.8 ± 0.1	3.9 ± 0.1	40	-
			0.3% bile salts	9.5 ± 0.1	9.5 ± 0.0	99	+
LAB67	*Lactococcus lactis*	Autochthonous	pH 2.5	7.0 ± 0.0	0.0 ± 0.0	0	-
			0.3% bile salts	6.6 ± 0.1	0.0 ± 0.0	0	-
LAB82	*Leuconostoc citreum*	Autochthonous	pH 2.5	9.5 ± 0.0	0.0 ± 0.0	0	-
			0.3% bile salts	0.0 ± 0.0	0.0 ± 0.0	0	-
LAB89	*Weisella cibaria*	Autochthonous	pH 2.5	6.2 ± 0.1	0.0 ± 0.0	0	-
			0.3% bile salts	7.7 ± 0.0	4.0 ± 0.0	53	-
LAB94	*Weissella confusa*	Autochthonous	pH 2.5	6.2 ± 0.4	1.4 ± 0.2	22	-
			0.3% bile salts	0.0 ± 0.0	0.0 ± 0.0	0	-
LAB97	*Lactiplantibacillus plantarum*	Autochthonous	pH 2.5	7.4 ± 0.0	7.5 ± 0.1	101	+
			0.3% bile salts	6.1 ± 0.3	6.9 ± 0.3	113	+
C244	*Lactiplantibacillus plantarum*	INIAV’s collection	pH 2.5	7.9 ± 0.1	4.2 ± 0.2	54	-
			0.3% bile salts	7.9 ± 0.0	6.1 ± 0.1	77	+
C1090	*Weissella paramesenteroides*	INIAV’s collection	pH 2.5	7.8 ± 0.1	9.5 ± 0.0	122	+
			0.3% bile salts	7.2 ± 0.2	6.8 ± 0.2	95	+

(+) LAB strain meets the minimal cell amount (10^7^ CFU/mL) to have probiotic potential; (-) LAB strain does not meet the minimal cell amount (10^7^ CFU/mL) to have probiotic potential; * (+) LAB strain meets the minimal cell amount (107 CFU/mL) to have probiotic potential; (-) LAB strain does not meet the minimal cell amount (107 CFU/mL) to have probiotic potential.

## Data Availability

Data will be made available upon reasonable request to the corresponding authors.

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
