# Peer review of "Selection of Autochthonous LAB Strains of Unripe Green Tomato towards the Production of Highly Nutritious Lacto-Fermented Ingredients"

_foods, 2021, doi:10.3390/foods10122916_

Round 1

Reviewer 1 Report

This manuscript describes selection of lactic acid bacteria with probiotic potential from unripe green tomato, as starter culture, to develop probiotic high-valued lacto-fermented products. In the world’s growing concern about food losses, fruits and vegetables represent a significant source of waste and by-products. Therefore, this study seems very meaningful to mitigate food losses in the fruit value chain.  The overall strategy of study and selection of strains had been properly conducted. However, it seems to be not enough in providing interesting information to readers. It seems that additional experiments is needed. It would be appropriate to describe the characteristics of product by fermenting unripe tomatoes using the strains selected in this study as a starter.

Author Response

Responses to reviewer 1 are given after the Reviewer comment, in blue.

Reviewer #1

This manuscript describes selection of lactic acid bacteria with probiotic potential from unripe green tomato, as starter culture, to develop probiotic high-valued lacto-fermented products. In the world’s growing concern about food losses, fruits and vegetables represent a significant source of waste and by-products. Therefore, this study seems very meaningful to mitigate food losses in the fruit value chain.  The overall strategy of study and selection of strains had been properly conducted. However, it seems to be not enough in providing interesting information to readers. It seems that additional experiments is needed. It would be appropriate to describe the characteristics of product by fermenting unripe tomatoes using the strains selected in this study as a starter.

We would like to thank the reviewer for the positive feedback on the thought-out strategy to mitigate food waste. The development of high-valued lacto-fermented products resulting from unripe green tomato fermentation using autochthonous LAB strains contributes to the development of bio-based solutions impacting the food industry on the value of commonly regarded tomato wastes in terms of nutrition, health, environmental benefits, and functionality, and in providing base information to develop new appealing food ingredients/products. This paper describes the first investigation on the diversity of LAB isolated from unripe green tomato (variety H1015) in Portugal’s industry tomato producing region. The H1015 variety is the most significant cultivar in the Ribatejo region of Portugal (municipalities of Azambuja, Benavente, Cartaxo, and Vila Franca de Xira), managed by the Competence Center for Industrial Tomato (CCTI). In this region, the amount of unripe green tomato annually discarded in the fields justifies its valorization. There is already a start-up (Green Fresh) very interested in valorizing these fruit by producing sauces based on unripe green tomatoes. Thusly, we were glad to hear the interest in the topic and how the selection process was easily understood.

The use of indigenous LAB strains to induce fermentation processes is increasing. In this regard, species isolated from the fruit will be the best starter cultures. Therefore, the strains adaptation to the matrix and the specific tomato production region (productive system) is assured. This selection study will also lead to more efficient fermentation of unripe green tomato and, ultimately, a fermented product with better sensory qualities. It is worth noting that the two strains from the INIAV collection were previously tested and yielded good results in unripe green tomato fermentation benefiting its use as starter cultures with probiotic properties (cf. section 2.3 and L204-206). Moreover, those LAB strains were also identified in the unripe green tomato microbiota (cf. section 3.1).

We acknowledge the need for further fermentation studies with the selected strains as fermentation starters. Such studies are already being considered; however, we aimed to highlight unripe green tomato fruit LAB diversity and select potential probiotic candidates as starter cultures in the present study. We believe that further information on these LAB will thus lead to the choice of strains with high fermentation production potential that can be used as starter cultures to improve the fermentation of immature green tomatoes.

Reviewer 2 Report

The document lacks focus and the title does not demonstrate the content of the document, because it is not clear which experiments demonstrate the use of LAB as a starter culture. The title mentions indigenous strains, although of those proposed by the authors for subsequent use in green tomato fermentation, only one is indigenous and the other belongs to the INIAV collection.

L103 Please specify the degree of ripeness of the tomato (1-6)
Justify the initial isolation of microorganisms on MRS medium only 
L123 Please specify the selection criteria for the seven molecularly characterized strains.
L171 Please correct Lactococcus
L194-194 Provide more information about the previously reported genera in spontaneously fermented tomato, tomato sauce and canned tomatoes.
L197-198 What selection criteria were used to choose between strains that presented the same genus and species?
I consider that Table 2 can be eliminated because the information on the molecularly characterized strains is included in Table 3, only indicate with an asterisk in Table 3 the strains that already belonged to the INIAV's collection.

The results in Table 3 do not represent phylogenetic relationships among the characterized microorganisms. The authors may modify the title of the table or, preferably, include the phylogenetic relationship study.

Does the column referring to the accession numbers correspond to the accession numbers of the annotated sequences of the present work? When reviewing this information, it does not coincide with what is reported in the present work.

Table 4 shows the survival percentage of the strains evaluated under different stress conditions, why are the values higher than 100 %? What is the criterion to consider them as probiotic? A strain cannot be considered as probiotic or potentially probiotic by its sole ability to survive conditions of simulated digestion, pH or salt resistance.

L280 please correct Lactobacillus
L300 Please argue what is the basis for your speculation, the ability to grow in solanine medium? What was the criterion for selection of the concentration evaluated?

Please review the updated taxonomy for Lactobacillus paracasei.

Author Response

Responses to reviewer 2 are given after the Reviewer comment, in blue.

Reviewer #2

The document lacks focus and the title does not demonstrate the content of the document, because it is not clear which experiments demonstrate the use of LAB as a starter culture. The title mentions indigenous strains, although of those proposed by the authors for subsequent use in green tomato fermentation, only one is indigenous and the other belongs to the INIAV collection.

We thank the reviewer for the comment; the use of unripe green tomato was already studied in a previous work (Pereira et al., 2021) in which it was proven its potential as a starter culture. However, we do agree that no fermentation trial was assayed in the present ms. and, consequently, we changed the title accordingly. The title now reads “Selection of autochthonous LAB strains of unripe green tomato towards the production of highly nutritious lacto-fermented ingredients”.

Reference:

Pereira, N.; Martins, P.; Gonçalves, E. M.; Ramos, A. C.; Abreu, M. Inoculated fermentation of immature-tomato with potential probiotic Lactiplantibacillus plantarum and Weissella paramesenteroides starter cultures. In Livro de Resumos do XV Encontro de Química dos Alimentos: Estratégias para a Excelência, Auntenticidade, Segurança e Sustentabilidade Alimentar, Funchal, Madeira, Portugal, September 5-8; Câmara, J. S., Pereira, J. A. M., Gouveia, R. P., Eds. Universidade da Madeira, Centro de Química da Madeira, Funchal, Portugal, 2021; Abstract number PC-A22, 238.

L103 Please specify the degree of ripeness of the tomato (1-6)

We added such information based on USDA color classification chart (L99-100).

Justify the initial isolation of microorganisms on MRS medium only.

We followed the standard procedure to isolate LAB strains (as described in ISO 15214, 1998), therefore using MRS for that purpose. In summary, green unripe tomato samples were blended with tryptone salt broth and serially diluted (up to 10-6) in tryptone salt broth. One millilitre of each dilution was pour-plated in de Man Rogosa and Sharpe (MRS) agar, and after incubation at 30°C for 72 hours, different individual colonies (5 to 10 colonies per dilution) were phenotypically selected. Selected colonies were further sub-cultured on MRS broth to obtain pure cultures and the stock cultures kept at -80 °C in MRS broth supplemented with a cryoprotectant (50% (v/v) glycerol) until further analysis. Isolation description was further complemented in the ms. (L106-114).

L123 Please specify the selection criteria for the seven molecularly characterised strains.

The selection criteria was identified in the updated L199-206 (“Further molecular identification was carried out in predominant LAB strains (>10% isolation frequency; Figure 1) and in the strains identified as Lactococcus lactis and Lactobacillus plantarum due to the widespread use in the food industry as probiotic microorganism and/or microbial starter [26].”); However, we do understand the reviewer concern and added such information also in the material and methods section (L122-125: “Seven autochthonous LAB strains from unripe green tomato were selected based on isolation frequency (>10%) and relevance for green tomato spontaneous fermentation (id.: LAB40, LAB49, LAB67, LAB82, LAB89, LAB94, and LAB97) and further identified by 16S rDNA gene sequencing.”).

L171 Please correct Lactococcus

We thank the reviewer and proceeded accordingly.

L194-194 Provide more information about the previously reported genera in spontaneously fermented tomato, tomato sauce and canned tomatoes.

We proceeded as suggested and added such information. L194-L198 now reads: “Some studies support the identified genera prevalence in tomato products such as spontaneously fermented green tomatoes (L. mesenteroides, L. casei, L. citreum, W. confusa, L. lactis and L. plantarum) [31,32], fresh tomato juice (L. plantarum, W. cibaria and W. confusa, L. brevis and P. pentosaceus) [17] and canned tomato (Leuconostoc sp., Pediococcus sp. e Lactobacillus sp.) [33].”

L197-198 What selection criteria were used to choose between strains that presented the same genus and species?

No specific criteria was employed, as it was a completely randomised choice since all isolates had high identification matches considering biochemical characterisation, namely fermentation patterns from API 50 CH. For instance, among isolates identified as Leuconostoc mesenteroides (LAB40, LAB41, LAB51, LAB53, LAB60, LAB69, LAB70, LAB79, LAB84, LAB86 and LAB88), the selected strain was randomly picked from those with an identification match of 99.9%. The same applies for all other selected strains.

As to clarify the selection, we added such information on L202-204: “Among isolates identified as of the same species, those with higher identification percentange were chosen. In cases where identification percentage matched between same species, isolates were randomly selected.”

I consider that Table 2 can be eliminated because the information on the molecularly characterised strains is included in Table 3, only indicate with an asterisk in Table 3 the strains that already belonged to the INIAV's collection.

We thank the suggestion and proceeded accordingly.  Table 2 was deleted and table 3 updated to table 3, indicating the INIAV's collection strains with an asterisk.

The results in Table 3 do not represent phylogenetic relationships among the characterised microorganisms. The authors may modify the title of the table or, preferably, include the phylogenetic relationship study.

We thank the suggestion and proceeded accordingly. Table 3 was updated to Table 2 and the title was changed to: “Suggested species of each consensus sequence BLASTed against NCBI nucleotide database” (L214-215).

Does the column referring to the accession numbers correspond to the accession numbers of the annotated sequences of the present work? When reviewing this information, it does not coincide with what is reported in the present work.

We clarify that the column referencing the accession numbers corresponds to the accession numbers retrieved from the BLAST program at NCBI (http://blast.ncbi.nlm.nih.gov/Blast.cgi) when checking for the consensus sequences obtained from sample sequencing generated data. However, we do not understand what the reviewer means with “When reviewing this information, it does not coincide with what is reported in the present work.”. We re-checked both the consensus sequences and accession numbers at BLAST (October 28, 2021) and always matched the information reported in the ms.

Table 4 shows the survival percentage of the strains evaluated under different stress conditions, why are the values higher than 100 %? What is the criterion to consider them as probiotic? A strain cannot be considered as probiotic or potentially probiotic by its sole ability to survive conditions of simulated digestion, pH or salt resistance.

The survival rate (%) values were expressed as the quotient of the initial Log10 LAB counts (N0), and the Log10 LAB counts after the incubation period (Nf). This information was stressed in the ms. (L153-154). As such, since some strains had final counts above the initial use inoculum, the ratio has above one and, therefore, the survival rate is >100%.

We argue that “LAB strains need to withstand hostile gastric conditions (low pH and presence of bile salts) and maintain a growth ≥ 7 log10 CFU/mL to have in vitro probiotic potential.” (L242-243). We realise that further testing should be carried out to prove strain’s probiotic value (e.g., antibiotic susceptibility, antimicrobial activity, and bacterial adherence); however, the found differential strain’s tolerance to low pH and bile salts can be considered as an early indicator of respective probiotic potential. As such, we added further text to clarify (L271-274).

L280 please correct Lactobacillus

We proceeded accondingly.

L300 Please argue what is the basis for your speculation, the ability to grow in solanine medium? What was the criterion for selection of the concentration evaluated?

Thank you for giving us the opportunity to further elaborate on this subject. LAB tolerance to solanine was evaluated since this glycoalkaloid might interfere with LAB growth (Friedman, 2002), compromising the fermentative process. On the other hand, given that some LAB strains degrade glycoalkaloids (e.g., Li et al., 2021), we hypothesise that this ability might primarily depend on LAB’s tolerance to glycoalkaloids. Knowing that the content of glycoalkaloids in unripe green tomato fruits can be high (Friedman, 2015), we set solanine concentration at 200 ppm based on Schrenk et al. (2020), foreseeing potential fermentation efficiency and lacto-fermented ingredients safety. As stated in the introduction section (L61-63), Schrenk et al. (2020) propose 200 ppm as a glycoalkaloids threshold concentration for human consumption.

References:

Li, C.; Kong, Q.; Mou, H.; Jiang, Y.; Du, Y.; Zhang, F. Biotransformation of alkylamides and alkaloids by lactic acid bacteria strains isolated from Zanthoxylum bungeanum meal. Bioresour. Technol., 2021, 330, 124944. https://doi.org/10.1016/j.biortech.2021.124944

Friedman, M. Chemistry and Anticarcinogenic Mechanisms of Glycoalkaloids Produced by Eggplants, Potatoes, and Tomatoes. J. Agric. Food Chem., 2015, 63, 3323–3337. https://doi.org/10.1021/acs.jafc.5b00818

Schrenk, D.; Bignami, M.; Bodin, L.; Chipman, J. K.; del Mazo, J.; Hogstrand, C.; Hoogenboom, L.; Leblanc, J. C.; Nebbia, C. S.; Nielsen, E.; Ntzani, E.; Petersen, A.; Sand, S.; Schwerdtle, T.; Vleminckx, C.; Wallace, H.; Brimer, L.; Cottrill, B.; Dusemund, B.; Grasl-Kraupp, B. Risk assessment of glycoalkaloids in feed and food, in particular in potatoes and pota-to-derived products. EFSA J., 2020, 18, 6222, 190 pp. https://doi.org/10.2903/j.efsa.2020.6222

Please review the updated taxonomy for Lactobacillus paracasei.

We thank the comment and proceeded accordingly; however, we point out to the fact that API-web software (API 50 CHL V5.1) retrieved match was for Lactobacillus paracasei (LAB61), as the database is yet to be updated for the current taxonomy.

Reviewer 3 Report

This manuscript mentioned the characterization of the unripe green tomato lactic microbiota to screen LAB strains for use as starter cultures in fermentation processes along with LAB strains available from INIAV’s collection. However, it is essential to review some important points:

Results and Discussion

Line 171. Change “lactococcus” by “Lactococcus”.

Line 271-272. Change “and colleagues” by “et al.”

Line 280. Change “lactobacillus” by “Lactobacillus”.

Figure 2. Improve image quality.

References

Check that all references present the following format for journal articles:

Author 1, A.B.; Author 2, C.D. Title of the article. Abbreviated Journal Name Year, Volume, page range.

Line 339, 360, 365, 369, 380, 390, 393, 397, 402, 416, and 433. It is not necessary to add the journal number.

Line 342 and 349. Add the page range.

Line 411. Delete year.

Author Response

Responses to reviewer 3 are given after the Reviewer comment, in blue.

Reviewer #3

This manuscript mentioned the characterization of the unripe green tomato lactic microbiota to screen LAB strains for use as starter cultures in fermentation processes along with LAB strains available from INIAV’s collection. However, it is essential to review some important points:

Results and Discussion

Line 171. Change “lactococcus” by “Lactococcus”.

We proceeded in accordance.

Line 271-272. Change “and colleagues” by “et al.”

We proceeded in accordance.

Line 280. Change “lactobacillus” by “Lactobacillus”.

We proceeded in accordance.

Figure 2. Improve image quality.

Figure 2 was formated according to journal recommendations.

References

Check that all references present the following format for journal articles:

Author 1, A.B.; Author 2, C.D. Title of the article. Abbreviated Journal Name YearVolume, page range.

Line 339, 360, 365, 369, 380, 390, 393, 397, 402, 416, and 433. It is not necessary to add the journal number.

Line 342 and 349. Add the page range.

Line 411. Delete year.

We reviewed all references in accordance; however, we point out that the given citation for the references identified to “add the page range” are correct, and no page range is indicated (electronic version only).

Round 2

Reviewer 1 Report

I think that the strategy to mitigate food waste by using lactic acid bacteria is very meaningful research. However, I think that the results described in the present study are not enough to be published in this journal.

Author Response

The authors respected the reviewer´s opinion. However, we are convinced of the interest of the study carried out according to the approach taken. We were therefore disappointed that we were not able to demonstrate to the reviewer its value for publication. For us, sharing this information is essential to developing knowledge and motivating further studies on autochthonous strains as starter cultures in fermented products. Especially for fermented by-products of plant origin, information on autochthonous cultures is scarce, particularly with probiotic potential versus the current use of commercial strains. The data in this manuscript allows the community a unique label of the selected strains that could be retrieved from the GenBank online servers, being available for further studies.

Reviewer 2 Report

Thank you for addressing the suggested revisions.

Authors stated: "However, we do not understand what the reviewer means by "In reviewing this information, it does not match what is reported in the present paper." We rechecked both the consensus sequences and accession numbers in BLAST (October 28, 2021) and they always matched the information reported in the ms."

Regarding this response, I meant that when sequencing of new isolates is performed, these sequences are deposited in the database (may be GenBank). Therefore, in the first review, when looking at the accession numbers, it was not clear whether they corresponded to the accession numbers of the sequences generated in the present work. 

The authors have clarified that these accession numbers refer to the consensus sequences compared and therefore the accession information does not correspond to that of the isolates of the present work. I am grateful to clarify this point, however it is necessary that the authors "annotate" their own sequences and provide the accession numbers provided. This is a very quick and simple process that can be covered during the review of the manuscript. 

Author Response

We gratefully thank the reviewer for the clarification, as well as the careful and insightful review of our manuscript. We proceeded as suggested, and our own sequences were submitted to the GenBank database (https://www.ncbi.nlm.nih.gov/genbank/). The provided accession numbers were updated on table 2 (L214), corresponding to the sequences generated in the present work, and the information will be available on November 13, 2021.
